# Recovery of Lac Resin from the Aqueous Effluent of Shellac Industry

Gaurav Badhani [1,2], Shruti Yadav [2], Elen Reji [2] and Subbarayappa Adimurthy [1,2,*]

1 Academy of Scientific & Innovative Research (AcSIR), Ghaziabad 201002, India
2 CSIR–Central Salt & Marine Chemicals Research Institute, Gautam Buddha Mar, Bhavnagar 364002, India
* Correspondence: adimurthy@csmcri.res.in

**Abstract:** Shellac and aleuritic acid manufacturing industries generate a lot of alkaline aqueous effluent during the process of manufacture of shellac and aleuritic acid from the seedlac/sticklac. The generated effluent contains lac resin, lac wax, lac dye and other water-soluble organic acids. Shellac industries in India face problems with the disposal of aqueous effluent due to the presence of considerable amounts of natural organic molecules and the dark colour solution. To address these problems, we have developed a novel method for the selective recovery of the lac resin from the alkaline aqueous effluent of shellac manufacturing industry. The recovered lac resin has been characterized by $^{13}$C-NMR, FT-IR and melting point and the data were compared with standard industrial-grade resin. The recovered lac resin was evaluated by the lac manufacturing industry for commercial applications.

**Keywords:** aqueous effluent; acidification; aleuritic acid; Alkali Hydrolysis; ethylacetate; lac resin; shellac





## 1. Introduction

Lac is a natural resin secreted by the tiny lac insects on certain host trees principally found in India, Thailand, China and Indonesia. Due to its nontoxic and biodegradable nature, it is used for various applications in pharmaceuticals, cosmetics and surface coatings industries [1,2]. Lac resin is a polyester complex of long chain polyhydroxy fatty acids and sesquiterpinic acids [3].

Raw lac or stick lac contains resin, wax, dye, insect debris, bark of host trees and some other impurities [3]. Hence, stick lac is refined to obtain purified products for commercial applications. The stick lac is scraped and processed (crushed and washed), it is washed with hot alkali solution to obtain a refined product known as seedlac [4]. Aleuritic acid is recovered from the seedlac by alkaline hydrolysis [5–8]. During the hydrolysis process, the by-products lac dye/resin and wax may partly dissolve and enter the aqueous effluent (filtrate). These by-products were recovered by various procedures. However, there were many difficulties faced by the industries to recover these by-products in their pure form in order to find commercial applications in the market. Presently, the shellac industries sell the aqueous effluent containing the dissolved organic components to adhesive industries at a reduced price, due to the lack of innovative technology to recover the valuable products from the alkaline aqueous effluent. Based on our green and sustainable processes [9–15], we envisioned herein the selective recovery of the lac resin from the alkaline aqueous effluent of the shellac and aleuritic acid manufacturing industry [16]. Accordingly, we developed a process for the recovery of lac resin and identified the suitable conditions [17] which are discussed in this manuscript.

The recovered lac resin has various applications: it has the ability to produce smooth surface, decorative and durable films from alcoholic solutions; it is an excellent adhesion and bonding material for a wide variety of surfaces, exhibiting high gloss and hardness; it has ultraviolet resistance of films and excellent compatibility with cellulosic materials, and

hence it is capable of producing laminated products [18]. The aim of the present work is to recover the pure lac resin from the alkaline aqueous effluent of seedlac, shellac, kiri lac and other lac processing industries.

The chemistry of lac resin was well investigated by Sukh Dev et al. in the 1970s [19–23]. According to Sukh Dev, the aldehydic acids are the primary acids of lac resin, and the corresponding alcoholic acid components arise from Cannizzaro-type reactions during hydrolysis by strong alkali [19]. The chemical structure of lac resin and its hydrolysed products are provided in Figure 1.

**Figure 1.** Chemical structure of lac resin and its hydrolysed products.

The naturally available lac resin is a brown-red in colour, which is due to the presence of natural dyes present in the seedlac. Lac dye is a mixture of at least five closely related laccaic acids derived from anthraquinone [24]. These five acids are laccaic acid A, B, C, D and E (Figure 2), all of which have an anthraquinoid structure with two carboxylic acid groups, except for laccaic acid D.

**Figure 2.** Structures of laccaic acids (natural dyes).

## 2. Materials and Methods

The raw material, alkaline aqueous effluent, was provided by the Jaiswal Shellac Industry, West Bengal-723143, India. All other laboratory chemicals (NaOH, KOH, HCl, $H_2SO_4$) and solvents (ethylacetate, diethyl ether, dichloromethane, dichloroethane, hexane and dimethyl acetamide) were procured from the local suppliers Loba Chemie Pvt Ltd. Mumbai, Maharashtra 400005 India and used as procured.

The alkaline aqueous effluent obtained from the Jaiswal Shellac Industry, West Bengal-723143, India, was in the form of a dark brown solution with pH = ~12.

Extraction of lac resin from alkaline aqueous effluent:

Into a 500 mL round bottomed flask, 100 mL of aqueous alkaline effluent (pH = ~12) was added and treated with dropwise addition of aqueous solution of $H_2SO_4$ (5–20% solution) to lower the pH of the solution to 1–7. To the acidified aqueous solution, 100 mL of organic solvent was added and the reaction mixture was stirred for 2–4 h, and allowed for the phase separation. Organic solvent was washed with water (100 mL) and separated the organic layer. From the organic layer, ethyl acetate was recovered (~75–80 mL) by

rotatory evaporator, the residue was transferred into a Petri dish, and dried in hot air oven at 60 °C for 4–12 h; the varying yields of the obtained lac resin depends upon the conditions. The recovered lac resin finds various commercial applications as discussed above.

Scale-up experiment:

In a 3.0 L round bottomed flask, 500 mL of alkaline aqueous effluent (pH = 12) was placed and treated with dropwise addition of aqueous solution of $H_2SO_4$ (750 mL of 5%) to obtain a pH of the solution of 1.0. To the acidified aqueous solution, 500 mL ethyl acetate was added and stirred well for 4 h with overhead stirrer and allowed for phase separation. The organic phase was separated through separating funnel and organic phase was washed with 500 mL of water. From the organic phase, 400 mL of ethyl acetate was recovered by rotatory evaporator. The residue was transferred in to a Petri dish, and dried on a hot plate at 80 °C (24 h) to obtain 60.0 g light brown lac resin. The lac resin recovery process from alkaline aqueous effluent is depicted in Figure 3.

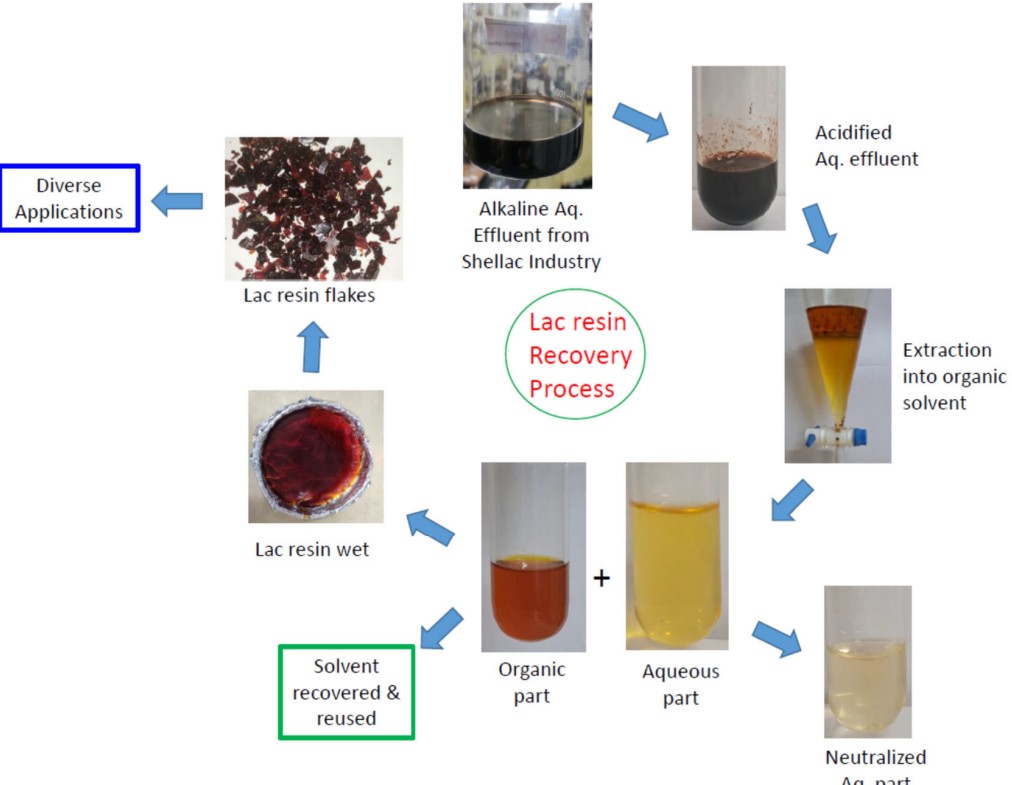

**Figure 3.** Schematic representation of lac resin recovery process from alkaline aqueous effluent.

The $^{13}$C-NMR spectra were recorded on 600 MHz (Jeol Resonance ECZ 600R spectrometer), FT-IR was recorded on Perkin Elmer Spectrum GX FT-IR system and melting points were determined with Thermo Scientific MEL TEMP instrument.

## 3. Results

The recovery of lac resin from the aqueous effluent depends upon many factors, such as the location of lac cultivation, season of lac cultivation, quality of seedlac and the amount of alkali used for washing of seedlac during the recovery process of shellac or aleuritic acid. The lac resin recovered with different solvents is presented in Table 1.

**Table 1.** Recovery of lac resin from the aqueous effluent [a].

| Sl. No. | Solvent | pH | Yield of Lac Resin (% wt/vol) | Nature of Lac Resin |
|---|---|---|---|---|
| 1 | Ethyl acetate | 7 (neutral) | 8 | Sticky dark brown solid |
| 2 | Ethyl acetate | 4.0 | 10 | Sticky brown solid |
| 3 | Ethyl acetate | 1.0 | 12 | Non-sticky light brown solid |
| 4 | Hexane | 4.0 | 1.0 | Brown sticky solid |
| 5 | Hexane | 1.0 | 1.2 | Brown sticky solid |
| 6 | Dichloromethane | 1.0 | 1.5 | Sticky light brown solid |
| 7 | Dichloroethane | 0.5 | 0.5 | Sticky grey solid |
| 8 | Dichloroethane | 1.0 | 2 | Sticky light brown solid |
| 9 | Diethyl ether | 1.0 | 3 | Non-sticky light brown solid |
| 10 | Dimethyl acetamide | 1.0 | 10 | Non-sticky dark brown solid |

[a] Conditions: 100 mL of aqueous solution, 100 mL of organic solvent, mixture was stirred for 4.0 h, the yield of lac resin reported after drying.

As can be seen from Table 1, initially, when 100 mL original alkaline aqueous solution (pH = 7) was treated with ethyl acetate, 8% yield of lac resin was obtained (Table 1, entry 1). When the same reaction was carried out at 4.0 pH, the lac resin yield was improved to 10% (Table 1, entry 2). Further lowering the pH of the solution, the yield was improved to 12% under the same conditions (Table 1, entry 3). To check the effect of lac resin recovery, various other organic solvents were screened, but no improvement in yield was observed (Table 1, entries 4-9). When the extraction of lac resin was carried out in hexane as solvent (at pH 4.0 and 1.0), very low yield was observed, due to the non-polar nature of the solvent (Table 1, entries 4 and 5). Further, when the extraction was performed in chlorinated solvents, such as dichloromethane and dichloroethane, with pH ranging from 0.5 to 1.0, no improvement in yield was observed (Table 1, entries 7–8). The experiments 6–8 indicate that the solubility of lac resin is poor in chlorinated solvents. The extraction of lac resin in diethyl ether provided only 3% yield (Table 1, entry 9). However, a comparative yield (10%) of lac resin was obtained with dimethyl acetamide as solvent (Table 1, entry 10). Though similar yields were obtained in ethyl acetate and dimethyl acetamide, from the perspective of economic viability, easy recoverability and environmental factors, ethyl acetate seem to be a better solvent, and hence scale-up experiments were performed with ethyl acetate. Experiment no. 3, Table 1, was scaled up to 500 mL of aqueous effluent and yield of lac resin was obtained, the detailed procedure is provided under experimental section.

## 4. Discussion

Better recovery of lac resin was observed with ethyl acetate as solvent with the pH of the aqueous solution at 1.0. The solvent used for the extraction of lac resin will be recovered and reused in the subsequent batches with the addition of the remaining fresh solvent, no discrepancies in yield were observed with the reused solvent. The solvent was recovered up to 80% during the course of the studies. It may be possible to improve the recovery of the solvent based on the temperature conditions and handling skills. After the recovery of lac resin from the aqueous effluent, the left out aqueous part could be neutralised with suitable bases and could be disposed easily. The aqueous part containing inorganic salts ($NaCl$ and $Na_2SO_4$) after the neutralisation may be considered for the recovery of salts by any conventional techniques. The recovered inorganic salts could be reused in the same process or find other possible applications, such as the leather and textile industry. The recovered lac resin was characterised by NMR, FT-IR and melting point, the data were compared with the standard industrial grade lac resin (Figure 4.).

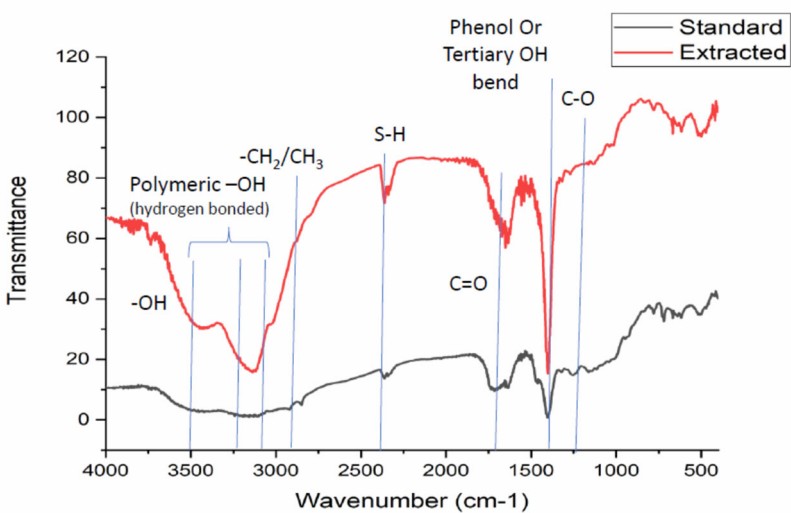

**Figure 4.** Comparison of FT−IR spectra of industrial standard lac resin and the lac resin recovered in the present study.

In the FT−IR spectra, the broad peaks observed between the 3200 and 3500 cm$^{-1}$ correspond to the hydrogen-bonded poly hydroxyl groups. The peaks around 2800 cm$^{-1}$ and 2400 cm$^{-1}$ are due to the presence of –CH$_2$-/CH$_3$ and SH groups. During the process of recovery of aleuritic acid, the mixture was treated with thioethanol, which remains in the aqueous effluent [5]. The broad bands at 1700 cm$^{-1}$ are due the presence of multiple carbonyl groups present as a mixture of aldehydes, carboxylic acids, esters and ketones as indicated in Figures 1 and 2. The sharp peak at 1400 cm$^{-1}$ is due to the presence of tertiary hydroxyl as well as phenolic groups. The lac resin is arranged to brown-red in colour due the presence of natural dyes anthraquinone derivatives [24]. The presence of these anthraquinones is also confirmed from $^{13}$C−NMR of both recovered and industrial-grade lac resin (see Supporting Information Figure S1). The C-O stretching bands around 1200–1300 cm$^{-1}$ tend to be less pronounced, and sometimes may overlap with another fingerprint region. The melting points of industrial and recovered lac resin are well matched, and both completely melt at 75–78 °C. From the $^3$C-NMR, spectra of both industrial-grade and recovered lac resin from the alkaline aqueous effluent were also recorded, and the spectral data are comparable. The delta values in the range of 120–150 ppm confirm the aromatic carbons and the value at 170–180 ppm for different aldehydic and carboxylic acid peaks (see Supporting Information Figure S1). Further, the applications of the recovered lac resin were explored by the industry in different forms and are provided in the supporting information (see Supporting Information Figure S2).

It may be noted that the amount of lac resin present in the alkaline aqueous effluent depends upon the shellac industry and also the dilution factor of the effluent during the recovery process of aleuritic acid. Further, the quality of lac resin depends upon the basic raw material (seedlac) used in the recovery process.

## 5. Conclusions

In conclusion, we have developed an efficient method for the selective recovery of lac resin form the alkaline aqueous effluent of the shellac industry. The best recovery of lac resin was observed with ethyl acetate as solvent system and by lowering the pH (<1.0) of the solution. As observed in the present study at the neutral pH of the solution, the recovery of the lac resin was poor. The solvent used in the process could be recovered and reused in the subsequent batches. The characterization data of recovered lac resin was compared with a standard industrial-grade sample and the data matched the commercial sample well. The recovery of lac resin depends upon several factors, such as dilution of original alkaline aqueous effluent generated by the shellac/aleuritic acid manufacturing industry, the quality of seedlac used and the location of the stick/seedlac cultivation. The

recovered lac resin was tested for various commercial applications by the shellac industry and finds good demand in the market.

## 6. Patents

Indian Patent Application No. 202111001134 (2020).

**Supplementary Materials:** The following supporting information can be downloaded at: https://www.mdpi.com/article/10.3390/suschem4010001/s1, Figure S1. $^{13}$C-NMR of (industrial grade & recovered) lac resin; Figure S2. Different forms of lac resin

**Author Contributions:** S.A.: Conceptualization of idea, supervision of the work, writing the manuscript. G.B.: Conducting experiments, collecting the data and the literature. S.Y.: Some experiments were carried out and validated. E.R.: Part of experiments were carried out. All authors have read and agreed to the published version of the manuscript.

**Funding:** We are thankful to CSIR-CSMCRI (MLP-027) for financial support.

**Institutional Review Board Statement:** Not applicable.

**Informed Consent Statement:** Not applicable.

**Data Availability Statement:** $^{13}$C-NMR spectra and different forms of lac resin.

**Acknowledgments:** CSIR-CSMCRI, Communication No. 215/2022. We are thankful to "Jaiswal Shellac Industry", West Bengal-723143 India, for providing the alkaline aqueous effluent for the study. G.B. is thankful to the University Grants Commission, New Delhi, India for providing Senior Research Fellowship to him.

**Conflicts of Interest:** The authors declare no conflict of interest.

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
