# Peer review of "Recovery of Lac Resin from the Aqueous Effluent of Shellac Industry"

_2673-4079, doi:10.3390/suschem4010001_

Round 1
Reviewer 1 Report
The authors present at fairly simple manuscript showing the optimization of lac resin recover from the aqueous effluent of the shellac industry. Considering the formula of the biopolymer, it was shown that ethyl acetate and pH adjustments could be used to increase the extracted yield of the resin... But it wasn't clear if the purity also was improved or preserved. It read very much like an undergraduate lab experiment, and it is unclear to this reviewer what the hypothesis was. In addition, what does the manuscript contain that the patent application referenced does not? The previous work by Dev, was extensive and other then using a rather predictable donor and hydrogen bond acceptor solvent for the extraction, I am not really sure what this manuscript was meant to prove or demonstrate.
Perhaps if the authors were do chromatography on the extract, it would add some detail as to the composition of the extract.
Author Response
Reviewer 1
Comments and Suggestions for Authors
Comment No 1. The authors present at fairly simple manuscript showing the optimization of lac resin recover from the aqueous effluent of the shellac industry. Considering the formula of the biopolymer, it was shown that ethyl acetate and pH adjustments could be used to increase the extracted yield of the resin... But it wasn't clear if the purity also was improved or preserved. It read very much like an undergraduate lab experiment, and it is unclear to this reviewer what the hypothesis was. In addition, what does the manuscript contain that the patent application referenced does not? The previous work by Dev, was extensive and other then using a rather predictable donor and hydrogen bond acceptor solvent for the extraction, I am not really sure what this manuscript was meant to prove or demonstrate.
Responses: We do agree with the reviewer comments that, by maintaining the pH of the solution at 1.0 or less than 1.0, the extraction efficiency was better with ethyl acetate as solvent. The ethyl acetate extract contains most of the resinous material which was partially hydrolysed during the alkali treatment and ethylacetate also contains the organic soluble part of the hydrolysed products. The recovered lac resin contains a composition partially hydrolysed resin along with a mixture of unidentified organic products. Due to these mixed composition, it retains the resinous properties useful for desired commercial applications. The manuscript contains the summarised information, precisely explaining the specific conditions useful for industrial applications for the selective extraction of lac resin from the aqueous effluent of aleuritic acid manufacturing industry. Whereas the patent covers a broader information in an elaborative manner. After, filing the patent, the same inventions/ideas described in a manuscript for broader visibility. The previous work reported by Sukh Dev and others are not suitable for industrial applications/practices as their objective was isolation of individual components by various separation techniques, not suitable for industrial scale applications.
Comment No 2. Perhaps if the authors were do chromatography on the extract, it would add some detail as to the composition of the extract.
Responses: We do agree with the reviewer comment, that to perform the column chromatography to identify the composition of the extract, but the objective of the present work is to identify the suitable solvent for the extraction of whole mass resin from the aqueous effluent of aleuritic acid manufacturing industry. Chromatographic separation procedure may not be applicable at industrial scale due to the cost factor of the process.
Reviewer 2 Report
1) Please tidy up the back matter. It says no conflicts of interest or funding was received, but then you name Jaiswal Shellac Industry as being partners. This creates some doubt. Funding is explained in the acknowledgements but not the funding section. I would remove lines 38-39 and state your objectives from a problem-solution perspective, rather than "we were asked to do it".
2) Line 95-96. This sentence is not appropriate for a methods section.
3) The results of Table 1 are not explained in sufficient detail. There is no characterization of the product. There are no applications of the lac resin despite alluding to 'diverse applications' so I cannot tell if the product can be used or not.
Author Response
Reviewer 2.
Comments and Suggestions for Authors
Comment No 1. Please tidy up the back matter. It says no conflicts of interest or funding was received, but then you name Jaiswal Shellac Industry as being partners. This creates some doubt. Funding is explained in the acknowledgements but not the funding section. I would remove lines 38-39 and state your objectives from a problem-solution perspective, rather than "we were asked to do it".
Responses: There is no conflicts of interest in the present work. No funding or grants received for this work. The Jaiswal Shellac Industry is acknowledged for providing the aqueous effluent required for the current study, and hence the industry has been acknowledged under the acknowledgement section. As suggested by the reviewer, we have removed the lines 38-39 and the sentence has been revised. The relevant reference no 6 also has been removed, and the remaining reference numbers are renamed from 6 onwards both in the text and in the reference section of the revised manuscript.
Comment No 2. Line 95-96. This sentence is not appropriate for a methods section.
Responses: We are thankful to the reviewer for the suggestion and we have removed these two lines (38-39) in the revised version of the manuscript.
Comment No 3. The results of Table 1 are not explained in sufficient detail. There is no characterization of the product. There are no applications of the lac resin despite alluding to 'diverse applications' so I cannot tell if the product can be used or not.
Responses: We are thankful to the reviewer for the constructive suggestions, accordingly the results of table 1 are discussed in details in the revised manuscript. The recovered product lac resin is not a single component, it is a mixture of partially hydrolysed products of seedlac, and hence the recovered product by the present procedure has been given to the industry for testing and evaluation for suitable commercial applications. As indicated by the shellac industry the recovered product (lac resin) is suitable to sell in the market. [However, the patented technology has been licensed to industry for commercial production].
Round 2
Reviewer 1 Report
see below
Author Response
In the rrevised manuscript, all the gramtical errors were corrected, and the changes made in the revised version are highlighted.
Reviewer 2 Report
This manuscript still lacks the necessary characterization to establish the composition and purity of the extract. If this is being performed by the industry partner that is fine but it needs to be reported. This could be pyrolysis-gas chromatography or NMR spectroscopy for example (compared to an authentic sample) or it could be material properties (strength, color, etc.). Thus the reviewer comments have not been fully addressed.
Author Response
Please see the attached letter for the responces.

Round 3
Reviewer 2 Report
Thank you for performing the requested characterization.